# Melatonin attenuates liver ischemia-reperfusion injury via inhibiting the PGAM5-mPTP pathway

Xiaoyi Shi[1,2,3☯], Jiakai Zhang[1,2,3☯], Jie Gao[1,2,3], Danfeng Guo[1,2,3], Shuijun Zhang[1,2,3], Xu Chen[1,2,3]*, Hongwei Tang[1,2,3]*

1 Department of Hepatobiliary and Pancreatic Surgery, The First Affiliated Hospital of Zhengzhou University, Zhengzhou, Henan, China, 2 Henan Engineering Technology Research Center of Organ Transplantation, Zhengzhou, Henan, China, 3 ZhengZhou Engineering Laboratory of Organ Transplantation Technique and Application, Zhengzhou, Henan, China

☯ These authors contributed equally to this work.
* tanghongwei301@163.com (HT); c7229996@163.com (XC)

**Data Availability Statement:** All relevant data are within the manuscript and its Supporting Information files.

## Abstract

Phosphoglycerate mutase/protein phosphatase (PGAM5)-mediated cell death plays an important role in multiple liver diseases. However, few studies have confirmed the regulatory mechanism of melatonin acting on PGAM5-mediated cell death in the context of liver ischemia-reperfusion (I/R) injury. The liver I/R injury model and cell hypoxia-reoxygenation model were established after melatonin pretreatment. Liver injury, cell activity, cell apoptosis, oxidative stress index, and PGAM5 protein expression were detected. To investigate the role of PGAM5 in melatonin-mediated liver protection during I/R injury, PGAM5 silencing, and overexpression were performed before melatonin pretreatment. Our results indicated that PGAM5 was significantly elevated by I/R injury, and predominantly localized in the necrosis area. However, treatment with melatonin blocked PGAM5 activation and conferred a survival advantage of hepatocytes in liver I/R injury, similar to the results achieved by silencing PGAM5. In terms of mechanism, we illustrated that activated PGAM5 promoted mitochondrial permeability transition pore (mPTP) opening, and administration of melatonin inhibited mPTP opening and interrupted hepatocytes death via blocking PGAM5. Our data indicated that the PGAM5-mPTP axis is responsible for I/R-induced liver injury. In contrast, melatonin supplementation blocked the PGAM5-mPTP axis and thus decreased cell death, providing a protective advantage to hepatocytes in I/R. These results established a new paradigm in melatonin-mediated hepatocyte protection under the burden of I/R attack.

## Introduction

Ischemia-reperfusion (I/R) is an inevitable process of liver surgeries, especially in liver transplantation [1]. I/R-related tissue injury is the prominent factor of organ rejection and liver dysfunction, accounting for about 10% of all early allograft failures [2]. Currently, progress has been made in research on intervention strategies for protecting liver I/R injury [3, 4]. Ischemic

**Funding:** This study was supported by the National Natural Science Foundation of China (Grants No.82372198), the Key Scientific Research Project of Henan Higher Education Institutions of China (Grants No.24B320024), Henan Sunshine Medical Health Development Foundation •Hepatobiliary Funding (Grants No. HKP2023003), and the Foundation of Henan Charity Federation (Grants No. GDXZ2023003).

**Competing interests:** The authors have declared that no competing interests exist.

**Fig 1. The chemical structure of melatonin.**

preconditioning is also a promising strategy for improving the outcome; however, the beneficial effect of ischemia preconditioning is limited to a portion of young patients with small liver resection volumes [5].

Although the pathogenesis of liver I/R injury is complicated, mitochondrial dysfunction appears to be a key factor [4]. Removing abnormal or dysfunctional mitochondria quickly not only preserves mitochondrial quality but also keeps cells and tissues alive [4]. Melatonin, which is a hormone secreted by the pineal gland, acts as an excellent antioxidant and can benefit a variety of health conditions, including diabetes, depression, infection, neurodegeneration, and metabolic syndrome [6]. (Fig 1) Because of its amphiphilic properties, melatonin can easily cross physiological barriers and enter cells [7]. Melatonin participates in many important physiological processes, including sleep regulation [8], cancer prevention [9], immune regulation [10], and circadian regulation [11]. Currently, the underlying mechanism of melatonin has been progressively unraveled, and mounting studies have revealed that melatonin can alleviate I/R injury by targeting autophagy [12, 13], inhibiting pyroptosis [14], alleviating inflammatory response [15], and blocking ROS-induced oxidation of DNA, proteins, and lipids [16]. As a powerful free radical scavenger, melatonin has been considered a potentially important biological antioxidant agent for numerous pathological conditions [6]. Melatonin protects against liver IR-induced mitochondrial dysfunction, and melatonin and its active metabolites protect cells from oxidative stress by regulating antioxidants and prooxidant enzymes [17]. Besides, melatonin also reduces mitochondrial swelling and glutamate dehydrogenase release, as well as acts on proteins associated with mitochondrial quality control [18]. Notably, melatonin has been reported to influence mitochondrial dysfunction in liver I/R injury in several studies, but many questions remain.

Recently, phosphoglycerate mutase family member 5 (PGAM5), a mitochondrial protein that belongs to the phosphoglycerate mutase (PGAM) family and possesses Ser/Thr phosphatase activity, was considered to play a key role not only in mitochondrial homeostasis but also in necrotic cell death [19]. Moreover, PGAM5 has been previously associated with various liver diseases, including acute liver injury [20], sepsis-induced liver injury [21], and liver cancer [22]. Hong found that Heme oxygenase-1 could alleviate liver I/R injury via PGAM5–mediated mitochondrial quality control [23]. Sun indicated that MiR-330-2p could inhibit PGAM5–induced mitophagy which alleviates hepatic IR injury in vivo [24]. Our previous study also found that ring finger protein (RNF)-5 protects the liver from I/R injury via mediating PGAM5 ubiquitination [25]. We also revealed that silencing of PGAM5 could mitigate liver I/R injury via inhibiting Drp1-mediated mitochondrial fission [26]. Given the crucial role of PGAM5 in liver I/R injury, taking effective measures to inhibit PGAM5 expression may become a viable treatment for liver I/R injury. Although the effects of melatonin on

mitochondrial dynamics have been widely investigated, information about the influence of melatonin on PGAM5 in liver I/R injury is still limited.

In this study, we indicated that melatonin exerts its liver-protective actions through MT2. Using PGAM5 silencing and overexpression strategies, we indicated that PGAM5 is essential in IR-mediated liver injury. Moreover, we found that mPTP opening is a critical step in the downstream processing of PGAM5-mediated liver I/R injury. Overall, we uncovered an essential role of the PGAM5-mPTP axis in the protective effect of melatonin against liver I/R injury.

## Materials and methods

### Cell culture and H/R

AML12 cells were got from Priella (Wuhan, China) and cultured in DMEM/F12 (Gibco, CA) containing 10% fetal bovine serum, 10 μg/ml Insulin, 5.5 μg/ml Transferrin, 5 ng/ml Selenium, and 40ng/ml Dexamethasone. Cells were maintained at 37°C with 5% $CO_2$ in a humidified incubator. To simulate an ischemic condition, the cells were incubated in a hypoxic incubator (Thermo Fisher Scientific) with 1% $O_2$ for the indicated time. To achieve reoxygenation, cells were cultured under normoxic conditions for 6 h [27]. For the pretreatment of AML12 cells with melatonin, the cells were treated with various concentrations of melatonin for 12 h, and then the cells were subjected to H/R as necessary.

### Measurements of cell activity/survival

Cell activity was measured by Cell Counting Kit-8 (Dojindo, Japan). Briefly, after treatment, 10ul of CCK8 solution was added to each well and subsequently incubated for 1 h at 37°C. Then, the plates were read at 450 nm by using a varioskan lux multimode microplate reader (Thermo Fisher Scientific). Cell survival was evaluated by Calcein/PI Cell Viability/Cytotoxicity Assay Kit (Biyuntian, China) and the TUNEL method (Servicebio, China) according to the manufacturer's instructions.

### Transfection

To overexpress PGAM5, AML12 cells were transfected with pLenti-CMV-PGAM5-GFP/Puro (Public Protein/Plasmid Library, China) using Lipofectamine®2000 transfection reagent (Invitrogen, Thermo Fisher Scientific, Inc.) according to the instructions. Following transfection, puromycin (2 μg/ml) was used to screen successful transfect cells and maintained for 2 weeks. The transfection efficiency was evaluated by western blotting.

### Mitochondrial membrane potential and ROS detection

Mitochondria membrane potential (MMP) was evaluated by JC-1 staining (Beyotime, China). In concisely, cells were washed 3 times with PBS and then incubated with JC-1 working solution (5 μg/ml) for 20 min at 37°C. Images were acquired by an inverted fluorescence microscope (Olympus IX71, Japan).

Intracellular ROS were measured by Dihydroethidium (DHE, Biyuntian, China). After washing with PBS, cells were stained with DHE at 37°C for 30 min. Images of DHE staining were visualized by an inverted fluorescence microscope (Olympus IX71, Japan).

### ATP test and mPTP opening rate determination

The level of ATP was detected by an ATP detection kit (Biyuntian, China). After H/R, the cells were lysed and the supernatants were incubated with the ATP working solution. Relative light units (RLU) were measured by chemiluminometer (Varioskan lux, Thermol Biotech, USA).

The ATP levels were calculated based on the standard curve, and the results were shown as nmol/mg protein.

The MPTP opening was evaluated via an MPTP detection kit (Biyuntian, China) in line with the instructions. After washing with PBS, the cells were cultured with calcein-AM/CoCl2 working buffer for 30min at 37˚C. Subsequently, the buffer was replaced with fresh prewarmed medium, and then the cells were further incubated for 30 min at 37˚C. The fluorescence was viewed under an inverted fluorescence microscope (Olympus IX71, Japan).

## Animal procedures and drug treatment

The study was carried out in compliance with the ARRIVE guidelines. Animal experiments were approved by the Ethics Committee of Zhengzhou University Hospital (Ethical number: KY-2021-00556) and cared for in accordance with the National Institutes of Health Guide for Laboratory Animals. Male C57BL/6 mice (8–10 weeks) were purchased from Vital River Laboratory Animal Technology Co., Ltd. (Beijing, China) and housed in our facility at an ambient temperature of 20–26˚C, humidity of 30–70% and 12-h dark/light cycle. To inactivate PGAM5 in C57BL/6 mice, adeno-associated virus (AAV) 8 system carrying scramble or shRNA against PGAM 5 (designed and synthesized by GeneChem, Shanghai, China) were injected into mice through tail vein 3 weeks before IR injury. The shRNA sequences used were as follows: CCAACTTCTCAGCTCAATTAA). Subsequently, a mice model of hepatic ischemia-reperfusion injury was constructed according to our previous study [25]. Briefly, the liver was exposed via laparotomy, and then, a micro-vessel clip (Fine Science Tools, cat.18055-06) was used to clamp the branches of the portal triad for 60 min followed by 6 h reperfusion. Melatonin (Sigma-Aldrich, 10 mg/kg, cat.73-31-4) was administered via intraperitoneal injection 15 min before ischemia and immediately after reperfusion [28]. After reperfusion for 6 h, all mice were euthanized under isoflurane anesthesia, and serum and liver samples were collected for future experiments. Then, the mice were randomly grouped (5 mice per group). (group 1) sham: injected with 0.9% saline; (group 2) I/R: inject with 0.9% saline and subjected to liver I/R; (group 3) I/R+MEL: inject with melatonin and subjected to liver I/R; (group 4) sham+vector: inject with AAV8 system carrying scramble; (group 5) sham+shPGAM5: inject with AAV8 system carrying shRNA against PGAM5; (group 6) I/R+vector: inject with AAV8 system carrying scramble and subjected to liver I/R; (group 7) I/R+shPGAM5: inject with AAV8 system carrying scramble and subjected to liver I/R.

## Immunofluorescence, and TUNEL

Liver tissues were fixed with 10% formalin, embedded in paraffin, and cut into 5 μm thick sections. HE staining was performed using the HE staining kit (Beyotime, C0105S, China) according to the manufacturer's instructions. For Immunofluorescence analysis, PGAM5 (PGAM5 (1:200, Abcam, ab126534) was used. Nuclei were counterstained with DAPI. TUNEL staining was performed using a Fluorescein (FITC) Tunel Cell Apoptosis Detection Kit (Servicebio, G1501, China) according to the manufacturer's instructions. At least three images were randomly taken by an inverted fluorescence microscope (Olympus IX71, Japan). for each stained slide, and at least three mice per group were tested. Image J software was used for image analysis of sections.

## Serum sample assays, SOD, and MDA detection

Serum ALT/AST levels were determined with a standard clinical automatic analyzer (Roche, Basel, Switzerland). The supernatant of liver homogenates was used for the measurements of SOD and MDA (Jiancheng Biochemical, China) according to the manufacturer's instructions.

**Table 1. Primers for qPCR.**

| Gene | Forward Prime | Reverse Prime |
|---|---|---|
| Complex-IV | 5′-CAGGATTCTTCTGAGCGTTCTATCA-3′ | 5′-AATTCCTGTTGGAGGTCAGCA-3′ |
| GAPDH | 5′-ACGGCAAATTCAACGGCACAGTCA-3′ | 5′-TGGGGGCATCGGCAGAAGG-3′ |

For the SOD assay, the plates were read at 560 nm by using a varioskan lux multimode microplate reader (Thermo Fisher Scientific). For the MDA assay, the absorbance was measured at 450, 532, and 600 nm by using a varioskan lux multimode microplate reader (Thermo Fisher Scientific).

## Western blotting

Liver tissue and cultured AML12 cells were homogenized by RIPA lysis (Biyuntian, China) containing protease and phosphatase inhibitors (PhosphoStop, Roche). Proteins were resolved by SDS-PAGE and then transferred to PVDF membranes. Membranes were then incubated with primary antibodies: Bcl-2 (1:1000, #3498, Cell Signaling), Bax (1:1000, #2772, Cell Signaling), cleaved- Caspase 3 (1:1000, #9661, Cell Signaling), β-tubulin (1:4000, Proteintech), PGAM5 (1:1000, ab308447, Abcam), MT1 (1:500, PA5-75749, ThermoFisher Scientific), MT2 (1:200, AMR-032- ThermoFisher Scientific), Caspase 3(1:2000, ab184787, Abcam). Horseradish peroxidase-conjugated anti-mouse(1:10000, SA00001-1, Proteintech) or anti-rabbit (1:10000, SA00001-2, Proteintech) was used as secondary antibodies. Proteins were detected using chemiluminescence (ECL, NCM Biotech) reagent, and the images were analyzed by Amersham Imager 800 (Amersham Biosciences, Buckinghamshire, UK).

## Real-time PCR

Total RNA was extracted from AML12 cells using TRIzol reagent (the CW Bio, Beijing, China) according to the manufacturer's instructions. Then the extracted RNA was reversely transcribed to cDNA applying a NeuScript II 1st strand cDNA synthesis kit (NUOWEIZAN, Nanjing, China). Quantitative real-time PCR was conducted by a 2 × SYBR Green PCR Master Mix (US EVERBRIGHT, Suzhou, China) in line with the manufacturer's protocol. Folding changes in gene expression were calculated through cycle threshold (Ct) values and normalized to the housekeeping gene (GAPDH). Primers are shown in Table 1.

## Statistical analysis

Statistical analysis was conducted using SPSS software (version 21.0). Measurement data were presented as the mean ± standard deviation (SD). The statistical significance was determined by the Student's t-test. Multiple group comparisons were analyzed using one-way ANOVA. A p-value of $<0.05$ was considered statistically significant.

## Results

### Melatonin alleviates liver I/R injury in mice

Serum ALT and AST levels were measured to assess liver damage. As shown in Fig 2A and 2B, we observed a sharp decrease in the levels of ALT and AST after pre-treatment with melatonin. Thereafter, HE-stained liver sections were evaluated for hepatic pathological changes. According to the morphological analysis, melatonin significantly reduced the area of necrosis tissue induced by I/R injury (Fig 2C). Similarly, TUNEL-positive cells were remarkably decreased in melatonin-pretreated mice (Fig 2D). Similar results were also obtained via the detection of

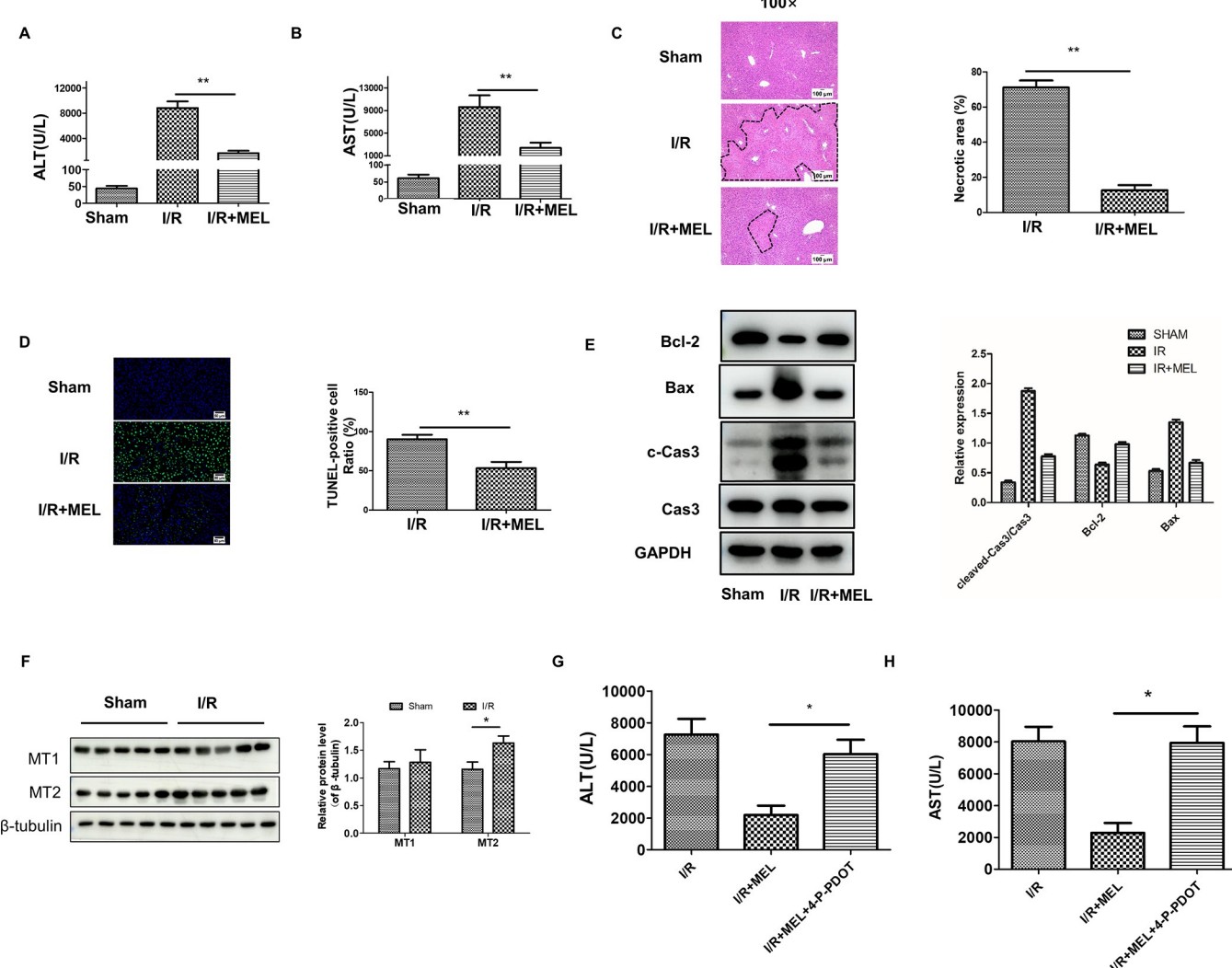

**Fig 2. Melatonin protects the liver against IR injury in mice.** Samples were collected after 1 h ischemia and 6 h reperfusion with or without melatonin. (A, B), The concentrations of ALT or AST were determined by a standard clinical automatic analyzer. (C), Representative images of H&E staining on mice liver sections (100X). (D), TUNEL assay was used to observe the hepatocyte death after liver I/R injury (200X), and the number of TUNEL-positive cells was analyzed. (E), Western blotting was used to measure the expression of Caspase 3, cleaved-caspase3, Bax, and Bcl-2 after I/R injury, and the relative protein expression was calculated. (F) Western blotting was used to measure the expression of MT1 and MT2 in liver of mice after I/R injury. (G, H) WT mice were intravenously injected with 4-P-PDOT (1mg/kg) or(and) melatonin (10mg/kg), the concentrations of ALT or AST were determined by a standard clinical automatic analyzer. n = 5 mice per group. *p<0.05, **p<0.01.

apoptosis-related proteins with western blotting. I/R injury treatment elevated the levels of cleaved caspase-3, and Bax, and decreased the level of Bcl-2, while melatonin supplementation could reverse these trends to some extent (Fig 2E). These results indicated that melatonin could alleviate liver I/R injury in mice.

To investigate the role of melatonin receptor, both MT1 and MT2 subtypes were detected in the liver of mice by western blot. Notably, MT2 expression significantly elevated after liver I/R injury, whereas MT1 level remained unaltered (Fig 2F). Afterward, we used 4P-PDOT to block MT2 receptors. As shown in Fig 2G and 2H, 4P-PDOT negated the effect of melatonin. These results indicated that melatonin exerts liver-protective actions through MT2.

## Melatonin attenuates hypoxia/re-oxygenation (H/R) injury in hepatocytes

The liver consists of many types of cells, but hepatocytes are the most abundant, making up more than 75% of the total liver volume in adults. Therefore, we hypothesized that the major target of melatonin was hepatocytes. Firstly, we evaluated the toxicity of melatonin in AML12 cells. As shown in Fig 3A, no significant effects were observed in normoxia. To mimic I/R injury in vitro, cultured AML12 cells were subjected to 12-h hypoxia and 6-h reoxygenation in the presence or absence of melatonin. As shown in Fig 3B, we observed sharply increased cell viability at the concentration of 100 μmol melatonin, which suggested that high-concentration melatonin produced hepatoprotection. Consequently, we used 100 μmol for a single concentration in subsequent experiments. Thus, Flow Cytometry analysis indicated that melatonin treatment significantly decreased AML12 cell apoptosis induced by H/R (I/R 61.8% vs. I/R +MEL 44.2%) (Fig 3C and 3D). Additionally, we performed a TUNEL experiment, which yielded similar results to flow cytometry (Fig 3E and 3F). These results indicated that melatonin could attenuate H/R injury in hepatocytes.

## Melatonin suppresses oxidative stress and alters PGAM 5 expression in hepatocytes

To investigate the relationship between liver IR injury and mitochondrial dysfunction, Tomm20 staining was used as an indicator of mitochondrial stress. Tomm20 staining revealed mitochondrial aggregates perinuclearly in mice subjected to liver I/R (Fig 3A). In contrast, IR-induced tomm20 augmentation was markedly decreased in melatonin-treated mice (Fig 4A). Oxidative stress was detected by the determination of MDA and SOD levels. As shown in Fig 4B, the concentrations of MDA were upregulated in the I/R group compared to the control group. Compared to the I/R group, the administration of melatonin remarkably ameliorated the level of MDA (Fig 4B). Similarly, the SOD level, an important antioxidant enzyme, was significantly decreased in the I/R group compared to the Sham group. And, melatonin reversed these changes (Fig 4C).

To further explore the mechanism of mitochondrial dysfunction during mice liver I/R injury, the alteration of PGAM5 was assessed, which plays a crucial role in mitochondrial dynamics. Western blot analysis showed that I/R-induced liver injury was associated with high expression of PGAM5 (Fig 4D). And, melatonin could inhibit the expression of PGAM5 (Fig 4D). Moreover, co-staining of PGAM5 with TUNEL showed that PGAM5 predominantly localized in the area of necrosis, indicating that PGAM5 might contribute to hepatocyte cell death (Fig 4E).

## Melatonin alleviates liver I/R injury by preventing PGAM5 upregulation

To investigate whether PGAM5 was involved in I/R-induced liver injury, we took advantage of mice infused with an adeno-associated virus that expressed scramble or shRNA against PGAM 5 (S1A Fig). Thus, the mice were exposed to liver I/R. Intriguingly, silencing of PGAM5 could alleviate liver I/R injury in mice as indicated by lower plasma levels of AST and ALT when compared with the control group (Fig 5A and 5B). Furthermore, histological analysis revealed that PGAM5-silenced mice displayed less liver tissue damage compared with the control mice (Fig 5C). Meanwhile, the number of TUNEL-positive cells in the liver from PGAM5-silenced mice was significantly decreased, compared with that of the control group (Fig 5D and 5E). Moreover, to investigate whether melatonin specifically inhibits PGAM5-mediated hepatocyte cell death, the PGAM 5—overexpression AML 12 cells were constructed by transfecting with lentiviral vectors (S1B Fig). To mimic I/R injury, AML12 cells were subsequently subjected to

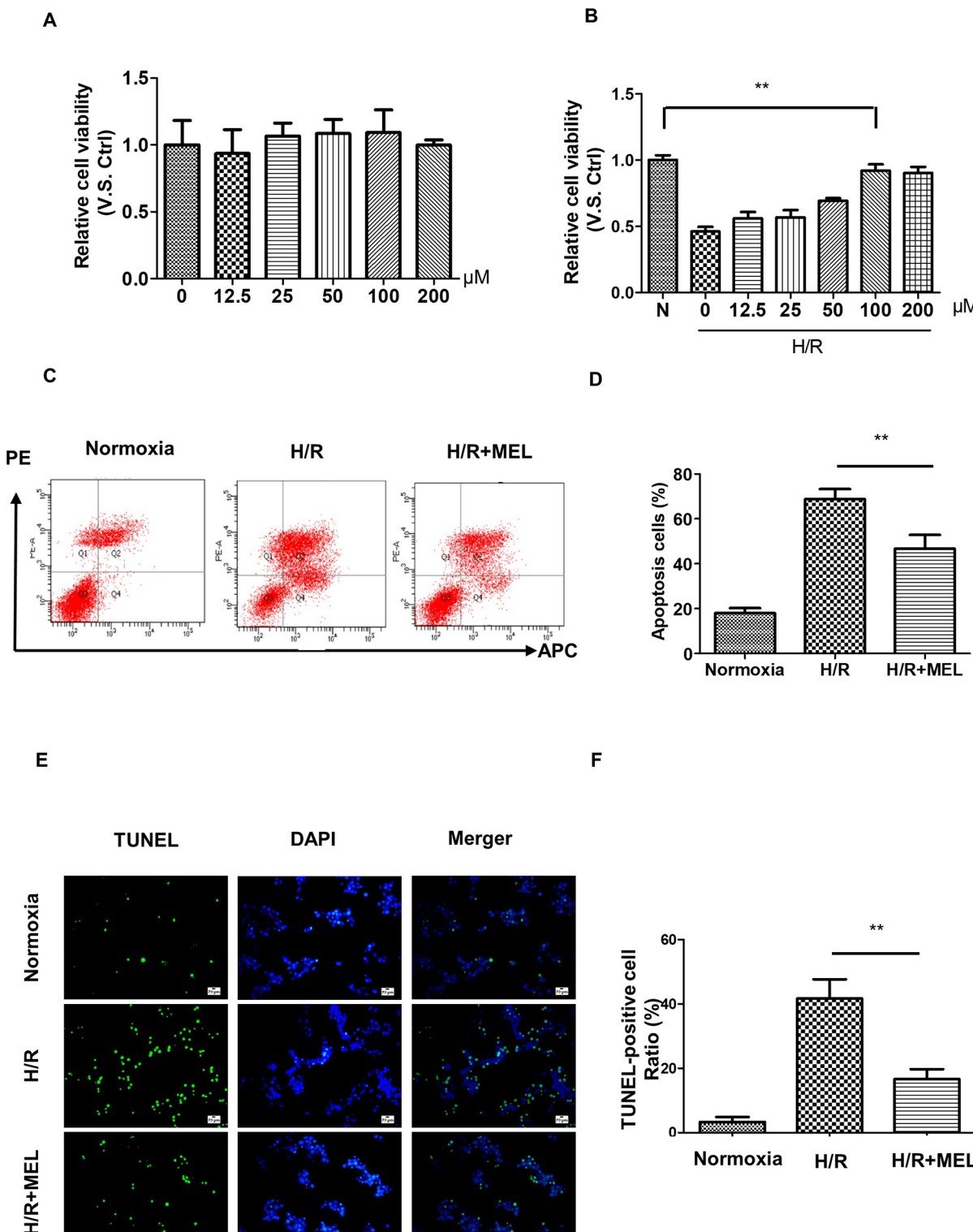

**Fig 3. Melatonin alleviates hepatocyte H/R injury.** (A), AML12 cells were treated with different melatonin concentrations under normoxic conditions. Cell viability was determined by CCK8 assay. (B), AML12 cells were subjected to 6 h hypoxia followed by 6 h of reoxygenation with or without melatonin. Cell viability was determined by CCK8 assay. (C, D), Apoptosis was detected by flow cytometry, and the ratio of dead cells was analyzed. (E, F), TUNEL assay was used to observe the dead cells after H/R injury (400X), and the number of TUNEL-positive cells was analyzed. **p<0.01.

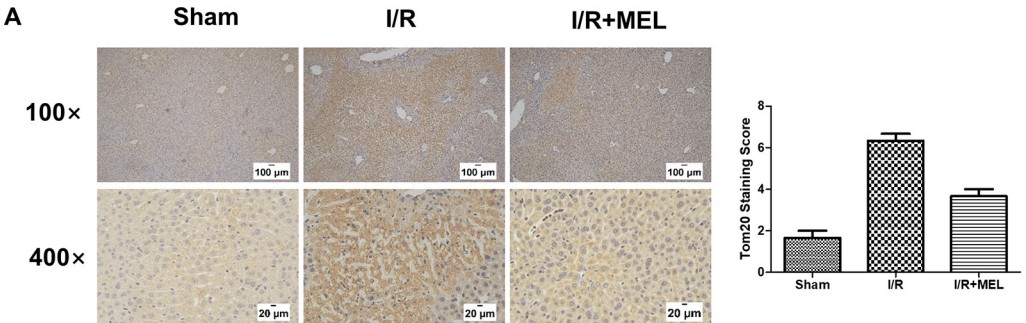

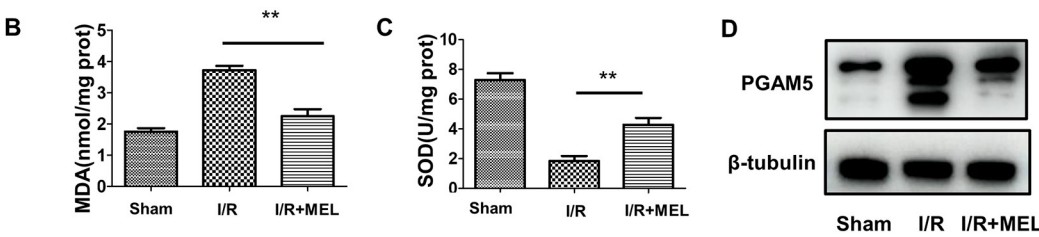

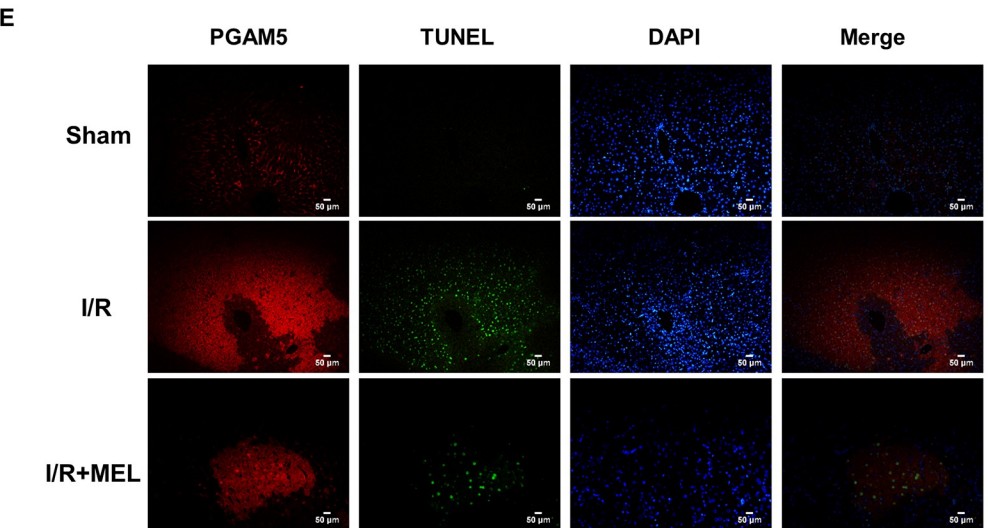

**Fig 4. I/R-induced Oxidative Stress and upregulation of PGAM5 are reversed by melatonin.** Samples were collected after 1 h ischemia and 6 h reperfusion with or without melatonin. (A). MDA in the supernatant of liver homogenates was analyzed. (B) SOD in the supernatant of liver homogenates was analyzed. (C, D). Expression of PGAM5 in liver tissues was analyzed by western blotting, and statistical analysis of protein expression. (E). Co-staining of PGAM5 (red) and TUNEL (green) in liver sections from mice (200X). n = 5 mice per group. * $p < 0.05$ vs Sham group; # $p < 0.05$ vs I/R group.

H/R. CCK8 assay indicated that overexpression of PGAM5 significantly reduced the cell viability and melatonin treatment did not appear to reverse the process (Fig 5F). Thus, Flow Cytometry analysis showed that overexpression of PGAM5 significantly increased the number of

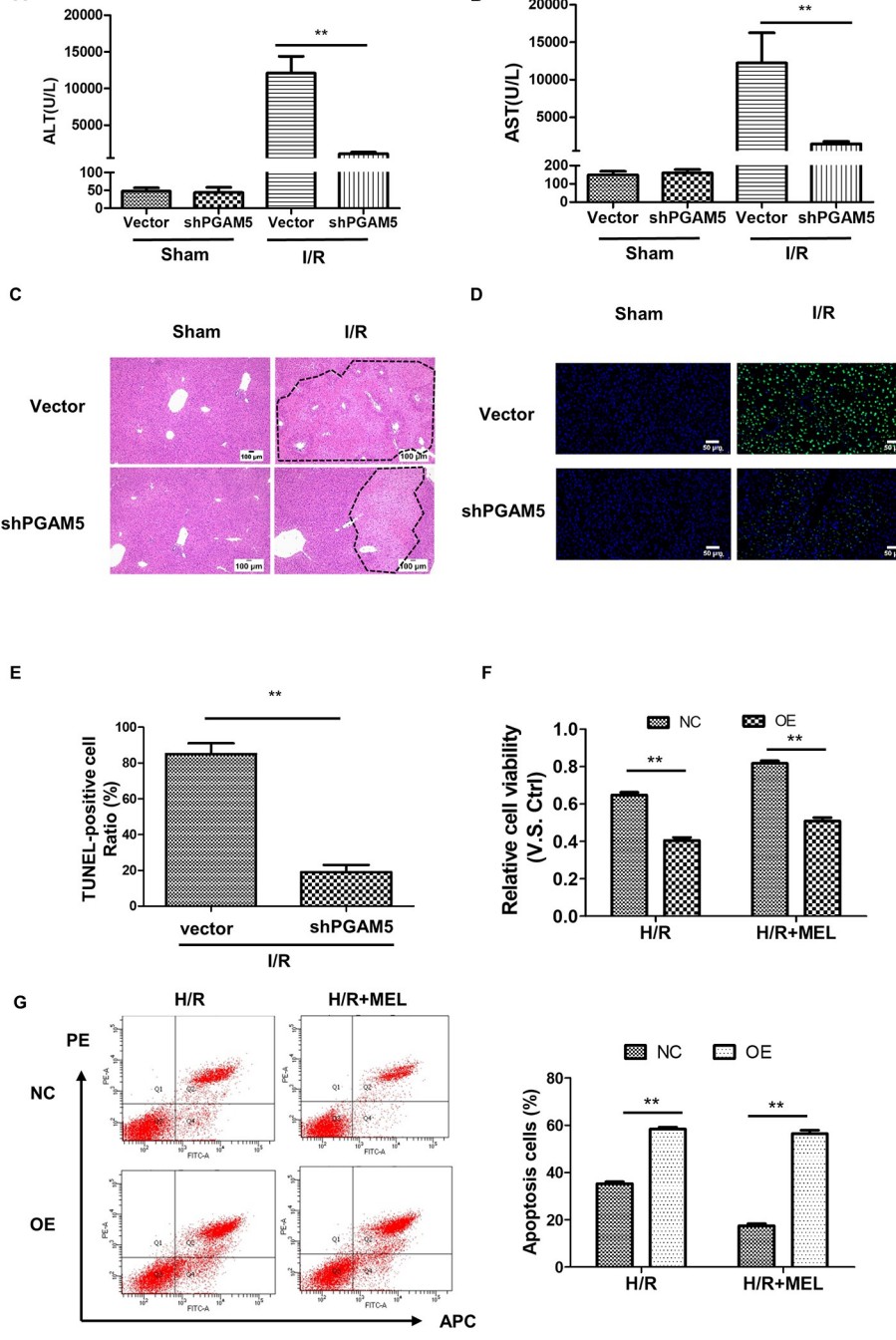

**Fig 5. Melatonin alleviates liver I/R injury in mice via inhibiting PGAM 5.** A-E. The PGAM5-silenced mice were used to confirm the role of PGAM5 in liver I/R injury. The levels of ALT (A) and AST (B) were analyzed by a standard clinical automatic analyzer. (C) Representative images of H&E staining on mice liver sections (100X). (D, E) Representative pictures of TUNEL staining and the number of TUNEL-positive cells were recorded. F, G. The PGAM 5-overexpression AML 12 cells were used to perform the gain-of-function analysis for PGAM5. (F) Cell viability was determined by CCK8 assay. (G) Apoptosis was detected by flow cytometry, and the ratio of dead cells was analyzed. n = 5 mice per group. **p<0.01.

dead cells induced by H/R and melatonin had no protective effect (Fig 5G). Collectively, these data demonstrated that PGAM5 is critical to hepatocyte cell death and organ pathology in I/R-induced liver injury.

## Melatonin maintains mitochondrial function by blocking PGAM5

Mitochondrial dysfunction contributes to the pathological process of liver I/R injury. Recently, PGAM5 has been identified as a crucial regulator of mitochondrial dysfunction. Thus, these findings prompted us to investigate the role of PGAM5 and mitochondrial function in melatonin-mediated protection of hepatocytes during liver I/R injury. As shown in Fig 6A, H/R-treated hepatocytes manifested a lower MMP, and such alteration could be reversed by

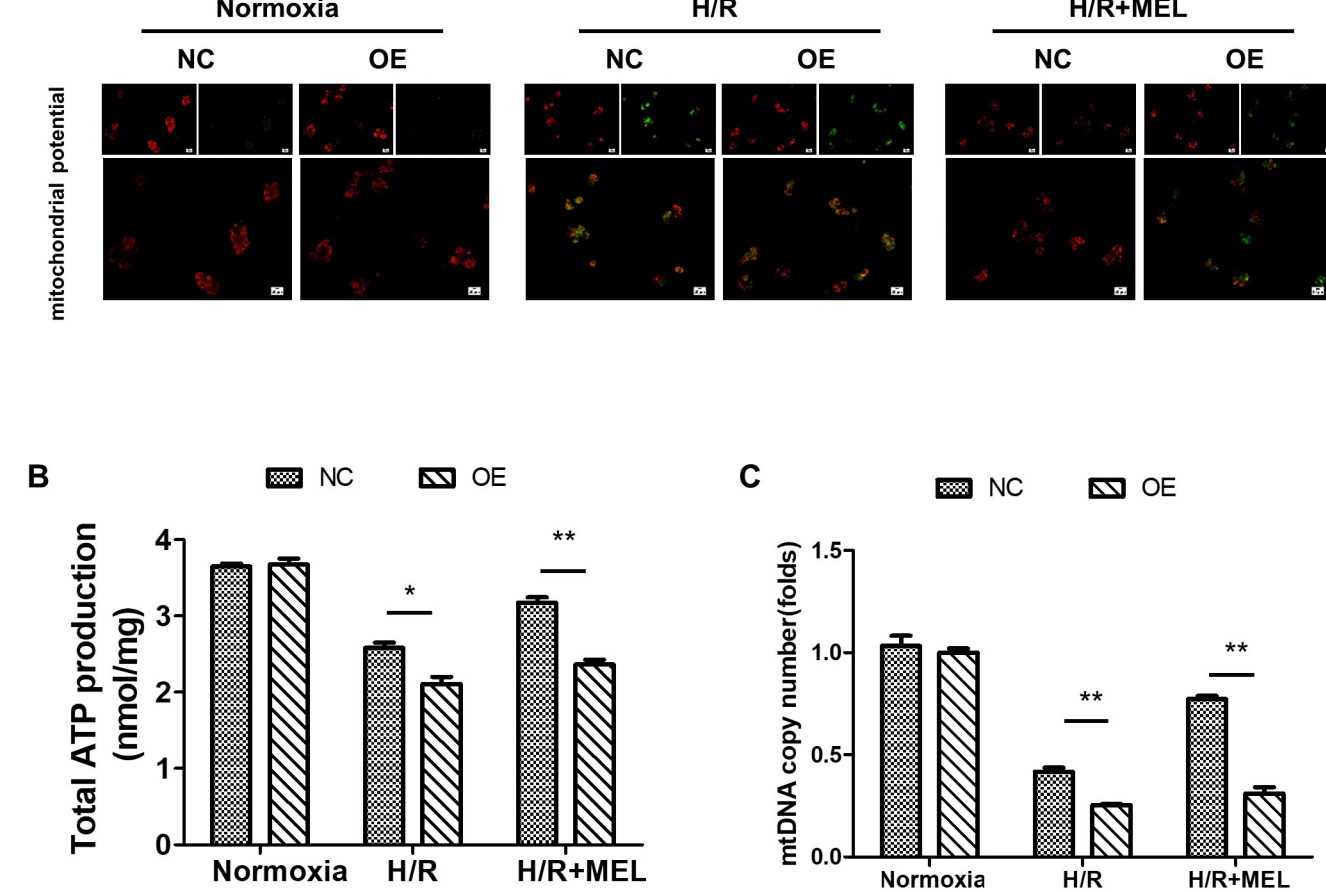

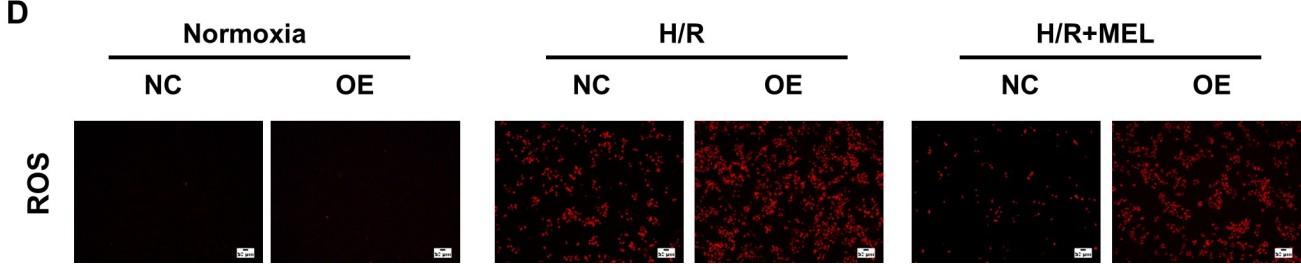

**Fig 6. Melatonin maintains mitochondrial function via repressing PGAM5.** (A) Representative images of H/R-treated AML12 cells loaded with the mitochondrial membrane potential indicator JC-1 (400 X). (B) The level of ATP production was measured in AML12 cells. (C) mtDNA copy number was assessed by complex IV segment. (D) Representative images of H/R-treated AML12 cells loaded with the mitochondrial ROS indicator DHE (200 X). $^{*}p < 0.05$, $^{**}p < 0.01$.

melatonin. However, the over-expression of PGAM5 counteracted the protective effects of melatonin (Fig 6A). Similarly, ATP synthesis (Fig 6B) and mtDNA copy number (Fig 6C) both were blocked by H/R injury but increased instead after melatonin treatment. However, PGAM5 overexpression negated the effect of melatonin. Moreover, following H/R, Intracellular ROS was significantly increased in PGAM5-overexpression AML12 cells but was maintained at a lower level after treatment with melatonin (Fig 6D). However, the overexpression of PGAM5 antagonized the inhibitory effect of melatonin on ROS production during H/R injury (Fig 6D). Collectively, our data indicated that PGAM5-mediated mitochondrial dysfunction is involved in hepatocytes cell death, and melatonin could block PGAM5 and thus maintain mitochondrial homeostasis.

## PGAM5 evokes hepatocytes cell death via promoting MPTP opening

MPTP opening is well known to regulate mitochondrial function and plays a crucial role in liver I/R injury. To further clarify the mechanism underlying melatonin-mediated PGAM5 inhibition and apoptosis inhibition in liver I/R injury, we measured the opening of mPTP. As shown in Fig 7A, the opening of mPTP was triggered by H/R but almost reversed to a near-normal level by melatonin. Nevertheless, the effect of melatonin was nullified by PGAM5 over-expression. These data demonstrated that PGAM5 overexpression enhanced mPTP opening. Afterward, Cyclosporin A (CsA), an inhibitor of mPTP opening, was used to verify whether mPTP opening is involved in PGAM5-mediated hepatocyte apoptosis. CCK8 assay showed that the viability of AML12 cells significantly increased after CsA or melatonin treatment (Fig 7B). Moreover, PGAM5 overexpression could not reverse the effect of CsA, which means that mPTP opening may be a downstream effector of PGAM5. To observe this phenomenon more intuitively, Calcein/PI staining was performed. As shown in Fig 7C, the H/R-mediated decreases in cellular survival rate were mostly reversed with CsA or melatonin. Furthermore, PGAM5 overexpression could not reverse the inhibition caused by CsA. Moreover, CsA also reversed the loss of MMP caused by H/R, similar to the effect of melatonin (Fig 7D). In contrast, CsA could also restore MMP even with high PGAM5 expression. Collectively, we concluded that mPTP opening is mainly regulated by PGAM5 and is implicated in liver I/R injury, whereas melatonin could block PGAM5-mediated mPTP opening and provide a survival advantage in liver I/R injury.

## Discussion

I/R injury is an unavoidable clinical problem for liver surgery. In clinical practice, liver I/R injury remains a key source of perioperative morbidity and mortality [29]. Due to the complex mechanisms involved, therapies to alleviate I/R remain limited at the bedside. In this study, we first examined the two melatonin receptor subtypes, MT1 and MT2 in the liver of mice, but MT2 was significantly increased in response to liver I/R injury. In addition, we indicated that melatonin could alleviate liver I/R injury and sustain mitochondrial function by inhibiting PGAM 5-related MPTP opening when it enters cells via MT2. Overall, in this study, we investigated the role of melatonin on PGAM 5 –related mitochondrial dysfunction in liver I/R injury. These findings fully confirmed the sufficiency of PGAM 5-related MPTP opening to mitigate liver I/R injury, and the necessity of melatonin to inhibit PGAM 5-mediated MPTP opening machinery.

The melatonin receptor is a G protein-coupled receptor that binds melatonin [30]. There have been three types of melatonin receptors identified to date. The MT1 and MT2 receptor subtypes are expressed in humans and other mammals, while MT3 mainly expressed in amphibia and birds [31]. In humans, mtnr1a is located on chromosome 4q35.2, while mtnr1b

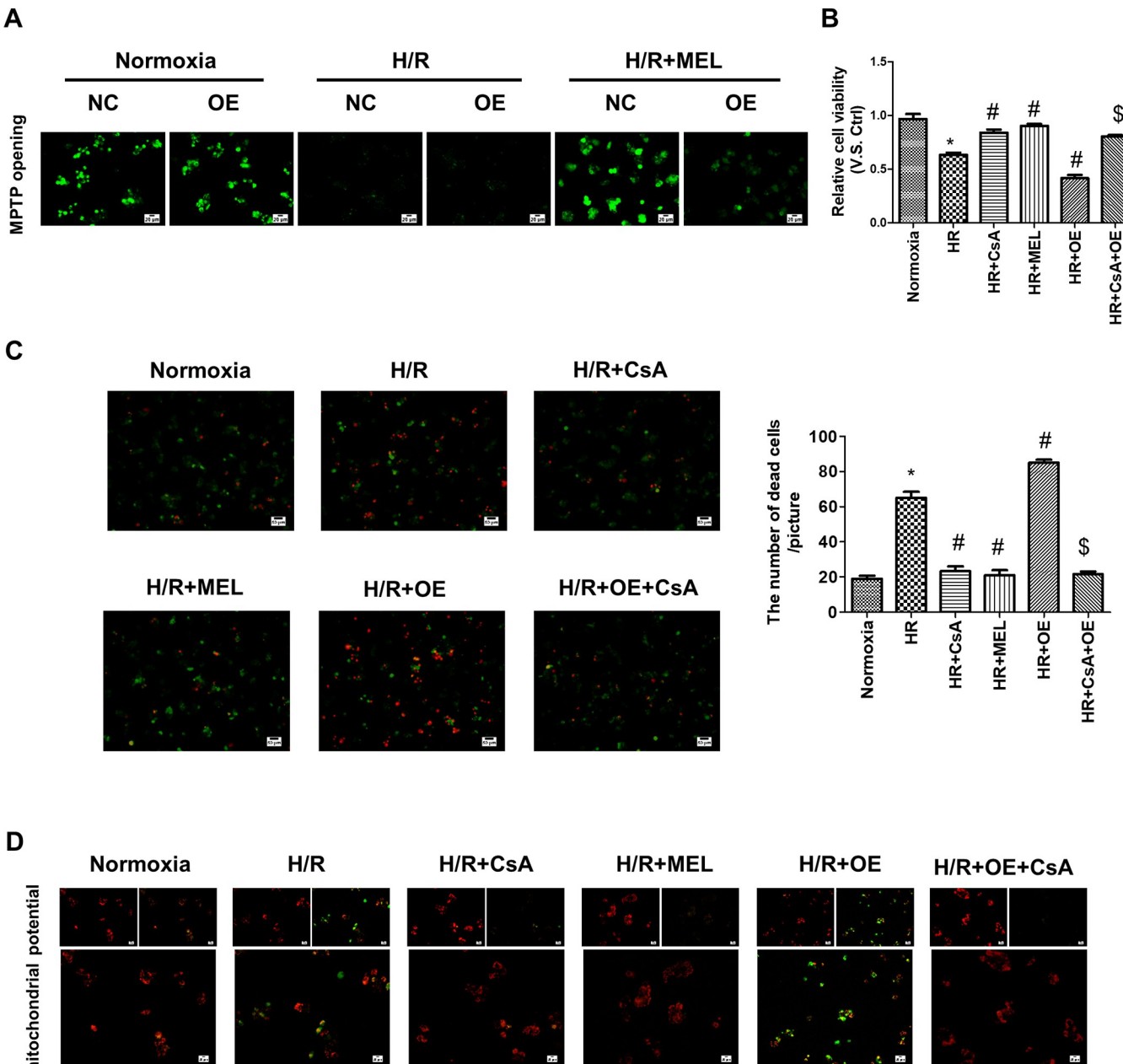

**Fig 7. PGAM5-mediated hepatocyte injury is mediated via mPTP opening.** (A) Representative images of mPTP opening in AML-12 cells after H/R injury (400X). (B) Cell viability was determined by CCK8 assay. CsA, an inhibitor of mPTP, was used to verify the role of mPTP opening in PGAM5-mediated hepatocyte injury. (C). Propidium iodide (PI) and calcein acetoxymethyl ester (calcein-AM) were used to evaluate the dead cells (200X). And the number of dead cells was calculated. (D) Representative images of JC-1 staining in H/R-treated AML12 cells (400X). * p<0.05 vs Normoxia group; # p<0.05 vs H/R group; $ p<0.05 vs H/R+OE group.

is located on chromosome 11q14. In previous reports, molecular characteristics of MT1 and MT2 have been well defined. In terms of molecular structure, pharmacological properties, and chromosomal location, MT1 and MT2 melatonin receptors differ greatly [32–34]. The MT1 are mostly present on pertussis toxin-sensitive Gi and inhibit forskolin-induced cAMP signaling and CREB phosphorylation. While the MT2 inhibits forskolin-evoked cAMP production and activates PKC in the suprachiasmatic nucleus, which might explain their unique

contributions to liver protection [35, 36]. In this study, we found that the expression of MT2 was significantly increased in the liver subjected to I/R injury. Using an MT2 antagonist P-PDOT negated the effect of melatonin, which means that melatonin exerts its liver-protective actions through MT2.

PGAM5 is a Serine/Threonine phosphatase that is mainly localized in the inner mitochondrial membrane [37]. Unlike the other members, PGAM5 lacks mutase activity. In humans, PGAM5 has two isoforms, PGAM5-L and PGAM5-S. The first 239 amino acids of both isoforms are the same. However, PGAM5-L has 50 extra amino acids and PGAM5-S has 16 extra hydrophobic amino acids [38]. PGAM5 participates in many cellular processes, including mitochondrial dynamics [37], mitophagy [39], programmed cell death [40], and the WNT/β-catenin pathway [41]. During I/R, mitochondrial dysfunction leads to a marked decrease in ATP synthesis, calcium overload, ROS surge, MMP loss, and mitochondrial membrane permeability increase 4]. Eventually, these factors contribute to impaired autophagy, necroptosis, and apoptosis of the cells [4]. As the intersection of mitophagy, necroptosis, and apoptosis, PGAM5 holds promise as a novel target for the treatment of I/R injury. In our previous study, we indicated that baseline deletion of PGAM5 could prevent liver I/R injury via inhibiting Drp1-mediated mitochondria fission in vivo and in vitro. In this study, we found that melatonin exerts its liver-protective actions by inhibiting PGAM5. the effects of melatonin were abolished in PGAM5-overexpression hepatocytes. Our data identify the injurious effects of PGAM5 in liver I/R injury. Moreover, melatonin appears to be a relatively effective agent against liver I/R injury via inhibiting the expression of PGAM5.

Multiple studies have demonstrated that mitochondrial dysfunction plays a crucial role in I/R injury [42, 43]. The timely removal of dysfunctional mitochondria not only sustains mitochondrial homeostasis but also contributes to the survival of cells. In this study, we found that melatonin failed to restore I/R-induced mitochondrial dysfunction in PGAM5-overexpression hepatocytes, which means that PAMG5-mediated mitochondrial dysfunction is involved in the protective effect of melatonin in liver I/R injury. Mechanistically, we found that the protective effect of melatonin is related to PGAM5-mediated MPTP opening. MPTP opening is mainly triggered by oxidative stress and calcium overload 32. Upon reperfusion, mitochondria release more free radicals. Excessive free radicals cause oxidative damage to mitochondrial respiratory chains, leading to more electron leakage and free radical production 33. Moreover, free radicals can increase mitochondrial permeability transition pore (MPTP) opening, resulting in loss of membrane potential. The opening of MPTP can trigger the release of calcium ions within the mitochondria and lead to mitochondrial dysfunction, thereby affecting mitochondria fission and fusion. During mitochondrial fission, the opening of MPTP may cause damage to the inner mitochondrial membrane and metabolic instability, and subsequent cytochrome C is released, which may exacerbate cellular stress and death. Upon release of cytochrome C into the extracellular medium, cells undergo caspase-dependent apoptosis [4]. The importance of MPT in I/R has been well demonstrated by the finding that CsA inhibits I/R-induced apoptosis, whereas tacrolimus, which does not affect MPT, could not alleviate I/R injury [44]. PGAM5-L facilitates the transfer of Keap1 into mitochondrial to sense the oxidation signal [45]. In our study, we indicated that PGAM5 inhibition applied by melatonin blocked the mPTP opening. The mPTP inhibitor CsA could inhibit apoptosis induced by H/R, even at PGAM5-overexpression cells. Consequently, PGAM5 may act as an important regulator of mPTP opening.

## Conclusions

Our data indicated that melatonin could alleviate I/R injury-mediated liver injury *in vivo* and in *vitro*. The exogenous application of melatonin blocks the expression of PGAM5 and

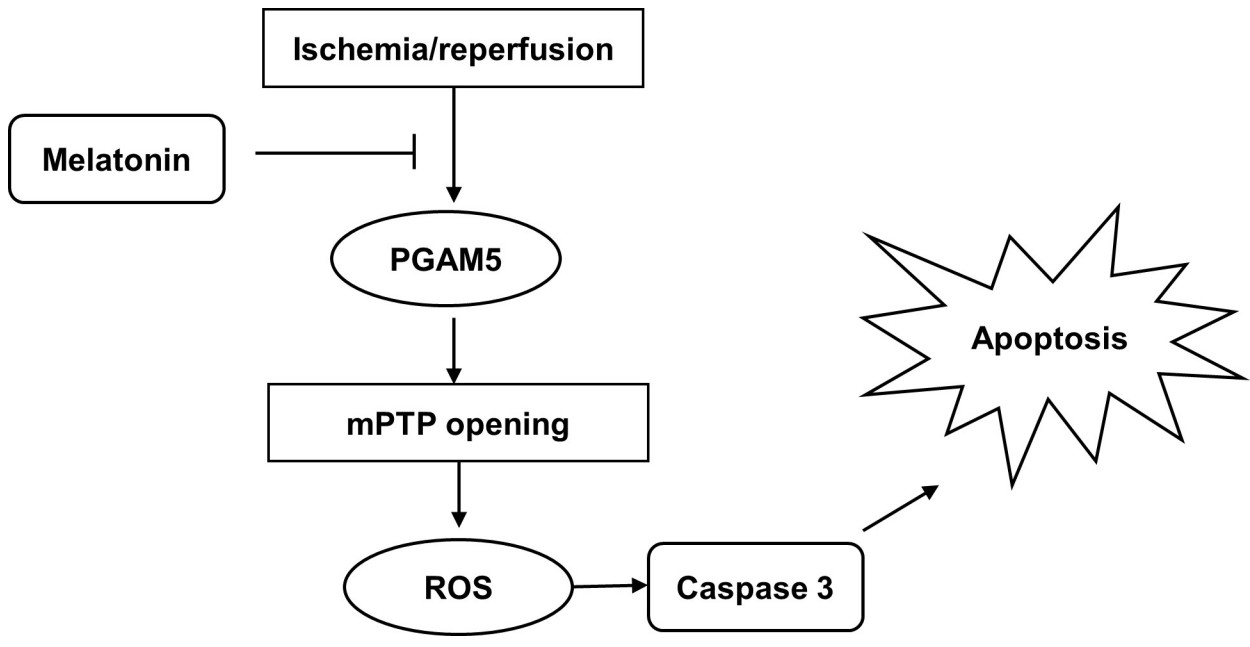

**Fig 8. A schematic of mechanisms of melatonin on liver protection in mice liver I/R injury.**

improves mitochondrial function. Our research still has several limitations. Firstly, PGAM5$^{fl/fl}$ mice seem to be a better alternative as a control group to exclude background differences. Secondly, the specific mechanism regarding PGAM5 regulation of mPTP opening needs to be further investigated (Fig 8).

## Supporting information

**S1 Fig. The silence efficiency and overexpression efficiency of PGAM5 expression were detected by western blotting analysis. (**A) Expression of PGAM5 in liver tissues from Vector and shPGAM5 mice was analyzed by western blotting. (B) the overexpression efficiency of PGAM5 in AML12 cells was verified by western blotting.
(TIF)

**S1 Raw data.**
(XLSX)

**S1 Raw images.**
(PDF)

## Author Contributions

**Conceptualization:** Xiaoyi Shi, Xu Chen.

**Data curation:** Xiaoyi Shi, Jiakai Zhang, Jie Gao, Danfeng Guo, Shuijun Zhang, Xu Chen.

**Funding acquisition:** Xiaoyi Shi, Jiakai Zhang, Danfeng Guo, Hongwei Tang.

**Investigation:** Xiaoyi Shi, Jiakai Zhang, Jie Gao, Danfeng Guo.

**Methodology:** Jiakai Zhang, Xu Chen.

**Supervision:** Xu Chen, Hongwei Tang.

**Validation:** Shuijun Zhang, Xu Chen, Hongwei Tang.

**Writing – original draft:** Hongwei Tang.

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
