## [Decision Letter · Decision Letter 0]

16 Jul 2024

PONE-D-24-18786Melatonin attenuates liver ischemia-reperfusion injury via inhibiting the PGAM5-mPTP pathwayPLOS ONE

Dear Dr. Tang,

Thank you for submitting your manuscript to PLOS ONE. After careful consideration, we feel that it has merit but does not fully meet PLOS ONE’s publication criteria as it currently stands. Therefore, we invite you to submit a revised version of the manuscript that addresses the points raised during the review process.

We look forward to receiving your revised manuscript.

Kind regards,

Michael Bader

Academic Editor

PLOS ONE

3. PLOS requires an ORCID iD for the corresponding author in Editorial Manager on papers submitted after December 6th, 2016. Please ensure that you have an ORCID iD and that it is validated in Editorial Manager. To do this, go to ‘Update my Information’ (in the upper left-hand corner of the main menu), and click on the Fetch/Validate link next to the ORCID field. This will take you to the ORCID site and allow you to create a new iD or authenticate a pre-existing iD in Editorial Manager. Please see the following video for instructions on linking an ORCID iD to your Editorial Manager account: https://www.youtube.com/watch?v=_xcclfuvtxQ".

4. To comply with PLOS ONE submissions requirements, in your Methods section, please provide additional information regarding the experiments involving animals and ensure you have included details on (1) methods of sacrifice, (2) methods of anesthesia and/or analgesia, and (3) efforts to alleviate suffering.

5. Please include your tables as part of your main manuscript and remove the individual files. Please note that supplementary tables (should remain/ be uploaded) as separate ""supporting information"" files. 

Reviewers' comments:

Reviewer's Responses to Questions

**Comments to the Author**

1. Is the manuscript technically sound, and do the data support the conclusions?

Reviewer #1: Yes

Reviewer #2: Yes

Reviewer #3: Partly

2. Has the statistical analysis been performed appropriately and rigorously? 

Reviewer #1: Yes

Reviewer #2: Yes

Reviewer #3: Yes

3. Have the authors made all data underlying the findings in their manuscript fully available?

Reviewer #1: Yes

Reviewer #2: Yes

Reviewer #3: Yes

4. Is the manuscript presented in an intelligible fashion and written in standard English?

Reviewer #1: Yes

Reviewer #2: Yes

Reviewer #3: Yes

5. Review Comments to the Author

Reviewer #1: The study entitled “Melatonin attenuates liver ischemia-reperfusion injury via inhibiting the PGAM5-mPTP pathway” has tried to examine the role of PGAM5 in hepatic IR injury and its involvement in the melatonin-mediated hepatic protection. The authors have used both in vivo and in vitro models to prove the hypothesis. Although the melatonin-induced protection in liver I/R injury has been well established via HO-1 and inhibiting the NF-κB signaling pathway, the role of PGAM-5 has been explored for the first time in this manuscript. In order to do so, the authors have overexpressed and silenced PGAM5 and extensive work has been done with western blots, immunofluorescence and immunohistochemistry analysis to support the protective effects of melatonin and PGAM5 silencing. The paper contains important and novel findings and should have implications in clinical settings dealing with diagnosing and treating liver injury

Reviewer #2: # Introduction

- PGAM5 (phosphoglycerate mutase 5) should be defined in the introduction.

- The results of recent relevant studies [1-5] have not been included in both the introduction and discussion sections. The introduction and discussion should be substantially revised to incorporate these findings.

#Methods

- The in vivo study duration is not well stated. The sentence "Following the experiment, all mice were euthanized..." does not clearly indicate the study duration.

- It is unclear why a group of IR+ShPGAM5+Melatonin was not included.

- The terminology should be consistent, using "concentration" for in vitro studies and "dose" for in vivo studies.

- It would be better to separate the in vitro and in vivo study methods, presenting the in vitro methods first, followed by the in vivo methods.

- Clarify whether the western blot and real-time PCR analyses were performed on cell samples or animal samples, and present the methods in the appropriate order.

- The abbreviation "mPTP" (mitochondrial permeability transition pore) should be defined upon first use.

#Results

- Specify the data expression format (mean±SD or mean±SEM).

- In the first paragraph of the results section, the phrase "in the same vein" should be revised.

- The duration of cell hypoxia induction should be indicated.

- The sentence "high dose melatonin produced hepatoprotection" should be rewritten.

- The descriptions of the results shown in Figure 4 are inaccurate and should be corrected.

- The quality of the immunofluorescence images in Figure 4 and other figures should be improved.

#Discussion

The discussion effectively summarizes the key findings and places them in the context of the existing literature. The proposed mechanism is clearly explained and supported by the experimental data. The introduction should also include an explanation of PGAM5, similar to the discussion.

- In the last paragraph of the discussion, the format of the references should be corrected.

1- Yang M, Wang Z, Xie J, Reyad-ul-Ferdous M, Li S, Song Y. Cyclophilin D as a potential therapeutic target of liver ischemia/reperfusion injury by mediating crosstalk between apoptosis and autophagy. Chronic Diseases and Translational Medicine. 2023 Sep 25;9(03):238-49.

2- Mohamed DZ, El AE, Sokar SS, Shebl AM, Abu-Risha SE. Targeting autophagy to modulate hepatic ischemia/reperfusion injury: A comparative study between octreotide and melatonin as autophagy modulators through AMPK/PI3K/AKT/mTOR/ULK1 and Keap1/Nrf2 signaling pathways in rats. European Journal of Pharmacology. 2021 Apr 15;897:173920.

3- El AE, Sokar SS, Shebl AM, Mohamed DZ, Abu-Risha SE. Octreotide and melatonin alleviate inflammasome-induced pyroptosis through inhibition of TLR4-NF-κB-NLRP3 pathway in hepatic ischemia/reperfusion injury. Toxicology and Applied Pharmacology. 2021 Jan 1;410:115340.

4- Gao Y, Li ZT, Jin L, Lin J, Fan ZL, Zeng Z, Huang HF. Melatonin attenuates hepatic ischemia-reperfusion injury in rats by inhibiting NF-κB signaling pathway. Hepatobiliary & Pancreatic Diseases International. 2021 Dec 1;20(6):551-60.

5- Mao B, Yuan W, Wu F, Yan Y, Wang B. Autophagy in hepatic ischemia–reperfusion injury. Cell death discovery. 2023 Apr 5;9(1):115.

Reviewer #3: The manuscript by Tang et al. shows that melatonin inhibits liver injury by reducing the activity PGAM5. I have two major comments:

1) You do not mention and cite your recent paper in Scientific Reports (PMID: 38609411) on PGAM5 and liver injury. Please discuss how the new findings relate with the published ones.

2) The most important question is the mechanism by which melatonin exerts its effects on hepatocytes. Do they have receptors and can you inhibit the effects by blocking them? What are the intracellular signalling pathways?

Minor:

3) Which tissue is shown in Fig. S1A? What happens in other organs of the mouse, is PGAM5 downregulated in the whole body? The legend for this figure needs to be more detailed.

4) Please check the axis labels in figure 5G: Death cells ...

6. PLOS authors have the option to publish the peer review history of their article (what does this mean?). If published, this will include your full peer review and any attached files.

Reviewer #1: No

Reviewer #2: No

Reviewer #3: No

---

## [Author Response · Author response to Decision Letter 0]

14 Aug 2024

Response to Reviewers' Comments

Melatonin attenuates liver ischemia-reperfusion injury via inhibiting the PGAM5-mPTP pathway （PONE-D-24-18786）

1. It appears that your ORCiD iD has not been validated in your Editorial Manager account and we are unable to proceed until that step is complete. 

To validate your ORCiD iD in Editorial Manager, please follow the steps below: 

a. In your Editorial Manager account, please go to ‘Update my Information’ (in the upper left-hand corner of the main menu), and click on the Fetch/Validate link next to the ORCiD field. 

b. This link will take you to the ORCiD site and allow you to create a new iD or authenticate a pre-existing iD in Editorial Manager. 

For additional instructions, please watch the following video for a step-by-step demonstration: https://www.youtube.com/watch?v=_xcclfuvtxQ

Explanation

Thanks very much! We have already validated our ORCiD id.

Explanation

Thanks very much! We have already revised it.

3. We note your current Data Availability statement is: "All relevant data are within the manuscript and its Supporting Information files."

However, in your manuscript your data availability stated as follows: "The datasets used and/or analyzed during the current study available from the corresponding author on reasonable request."

Please update your Data Availability statement accordingly.

Explanation

Thanks very much! We have already revised it.

4. To comply with PLOS ONE submissions requirements, please provide the following information in the Methods section of the manuscript.

If anesthesia, euthanasia, or any kind of animal sacrifice is part of the study, please include briefly in your statement which substances and/or methods were applied.

Explanation

Thanks very much! We have already revised it.

Reviewer Comments:

Reviewer #1: The study entitled “Melatonin attenuates liver ischemia-reperfusion injury via inhibiting the PGAM5-mPTP pathway” has tried to examine the role of PGAM5 in hepatic IR injury and its involvement in the melatonin-mediated hepatic protection. The authors have used both in vivo and in vitro models to prove the hypothesis. Although the melatonin-induced protection in liver I/R injury has been well established via HO-1 and inhibiting the NF-κB signaling pathway, the role of PGAM-5 has been explored for the first time in this manuscript. In order to do so, the authors have overexpressed and silenced PGAM5 and extensive work has been done with western blots, immunofluorescence and immunohistochemistry analysis to support the protective effects of melatonin and PGAM5 silencing. The paper contains important and novel findings and should have implications in clinical settings dealing with diagnosing and treating liver injury。

Explanation

Thanks very much!

Reviewer #2: # Introduction

- PGAM5 (phosphoglycerate mutase 5) should be defined in the introduction.

Explanation

Thanks very much! We have already defined it in the revised manuscript.

- The results of recent relevant studies [1-5] have not been included in both the introduction and discussion sections. The introduction and discussion should be substantially revised to incorporate these findings.

Explanation

Thanks for your suggestion. We have already revised it. 

#Methods

- The in vivo study duration is not well stated. The sentence "Following the experiment, all mice were euthanized..." does not clearly indicate the study duration.

Explanation

Thanks for your suggestion. We have already revised it. 

- It is unclear why a group of IR+ShPGAM5+Melatonin was not included.

Explanation

Thanks for your suggestion. Our previous results have shown that melatonin can inhibit PGAM5 expression. Therefore, we thought that if silencing PGAM5 could reduce mouse liver IRI, then Melatonin can work by inhibiting PGAM5. Moreover, our preliminary experiments also found that melatonin had no effect in PGAM5-silenced mice (data not shown). In the recovery experiment, we selected aml12 cells and transfected PGAM5 with lentivirus to see whether overexpression of PGAM5 could neutralize the role of melatonin. We think that these two positive and negative experiments can clarify our hypothesis. Therefore, we did not set up the IR+ShPGAM5+Melatonin group. 

- The terminology should be consistent, using "concentration" for in vitro studies and "dose" for in vivo studies.

Explanation

Thanks for your suggestion. We have already revised it. 

- It would be better to separate the in vitro and in vivo study methods, presenting the in vitro methods first, followed by the in vivo methods.

Explanation

Thanks for your suggestion. We have already revised it. 

- Clarify whether the western blot and real-time PCR analyses were performed on cell samples or animal samples, and present the methods in the appropriate order.

Explanation

Thanks for your suggestion. We have already revised it. 

- The abbreviation "mPTP" (mitochondrial permeability transition pore) should be defined upon first use.

Explanation

Thanks for your suggestion. We have already revised it. 

#Results

- Specify the data expression format (mean±SD or mean±SEM).

Explanation

Thanks for your suggestion. We have already revised it. 

- In the first paragraph of the results section, the phrase "in the same vein" should be revised.

Explanation

Thanks for your suggestion. We have already revised it. 

- The duration of cell hypoxia induction should be indicated.

Explanation

Thanks for your suggestion. We have already indicated the duration of cell hypoxia in the revised manuscript. 

- The sentence "high dose melatonin produced hepatoprotection" should be rewritten.

Explanation

Thanks for your suggestion. We have already revised it. 

- The descriptions of the results shown in Figure 4 are inaccurate and should be corrected.

Explanation

Thanks for your suggestion. We have already revised it. 

- The quality of the immunofluorescence images in Figure 4 and other figures should be improved.

Explanation

Thanks for your suggestion. The figures have been improved in the revised manuscript.

#Discussion

The discussion effectively summarizes the key findings and places them in the context of the existing literature. The proposed mechanism is clearly explained and supported by the experimental data. The introduction should also include an explanation of PGAM5, similar to the discussion.

Explanation

Thanks for your suggestion. We have already revised it. 

- In the last paragraph of the discussion, the format of the references should be corrected.

Explanation

Thanks for your suggestion. We have already revised it. 

1- Yang M, Wang Z, Xie J, Reyad-ul-Ferdous M, Li S, Song Y. Cyclophilin D as a potential therapeutic target of liver ischemia/reperfusion injury by mediating crosstalk between apoptosis and autophagy. Chronic Diseases and Translational Medicine. 2023 Sep 25;9(03):238-49.

2- Mohamed DZ, El AE, Sokar SS, Shebl AM, Abu-Risha SE. Targeting autophagy to modulate hepatic ischemia/reperfusion injury: A comparative study between octreotide and melatonin as autophagy modulators through AMPK/PI3K/AKT/mTOR/ULK1 and Keap1/Nrf2 signaling pathways in rats. European Journal of Pharmacology. 2021 Apr 15;897:173920.

3- El AE, Sokar SS, Shebl AM, Mohamed DZ, Abu-Risha SE. Octreotide and melatonin alleviate inflammasome-induced pyroptosis through inhibition of TLR4-NF-κB-NLRP3 pathway in hepatic ischemia/reperfusion injury. Toxicology and Applied Pharmacology. 2021 Jan 1;410:115340.

4- Gao Y, Li ZT, Jin L, Lin J, Fan ZL, Zeng Z, Huang HF. Melatonin attenuates hepatic ischemia-reperfusion injury in rats by inhibiting NF-κB signaling pathway. Hepatobiliary & Pancreatic Diseases International. 2021 Dec 1;20(6):551-60.

5- Mao B, Yuan W, Wu F, Yan Y, Wang B. Autophagy in hepatic ischemia–reperfusion injury. Cell death discovery. 2023 Apr 5;9(1):115.

Reviewer #3: The manuscript by Tang et al. shows that melatonin inhibits liver injury by reducing the activity PGAM5. I have two major comments:

1) You do not mention and cite your recent paper in Scientific Reports (PMID: 38609411) on PGAM5 and liver injury. Please discuss how the new findings relate with the published ones.

Explanation

Thanks for your suggestion. When we discovered the role of PGAM5 in liver I/R injury in mice, we wondered if we could find a small-molecule inhibitor that could inhibit PGAM5. Our review of the literature found that Melatonin protects mitochondrial function and reduces myocardial ischemia-reperfusion damage. Therefore, we speculate that melatonin may reduce liver I/R injury by regulating hepatocyte mitochondrial function through PGAM5. Since the two projects were completed almost simultaneously, at the time of writing this article, the other one had not yet been published. Therefore, we did not cite this article. 

2) The most important question is the mechanism by which melatonin exerts its effects on hepatocytes. Do they have receptors and can you inhibit the effects by blocking them? What are the intracellular signalling pathways?

Explanation

Thanks for your suggestion. The literature has confirmed that melatonin activates two high-affinity G protein-coupled receptors, termed MT1 and MT2, to exert beneficial actions. Therefore, we did not further investigate this. In this study, we mainly explored how melatonin, after entering cells, regulates mitochondrial function and alleviates liver IRI in mice. 

1.Liu J, Clough SJ, Hutchinson AJ, Adamah-Biassi EB, Popovska-Gorevski M, Dubocovich ML. MT1 and MT2 Melatonin Receptors: A Therapeutic Perspective. Annu Rev Pharmacol Toxicol. 2016;56:361-83. doi: 10.1146/annurev-pharmtox-010814-124742. Epub 2015 Oct 23. PMID: 26514204; PMCID: PMC5091650.

2. Kinker GS, Ostrowski LH, Ribeiro PAC, Chanoch R, Muxel SM, Tirosh I, Spadoni G, Rivara S, Martins VR, Santos TG, Markus RP, Fernandes PACM. MT1 and MT2 melatonin receptors play opposite roles in brain cancer progression. J Mol Med (Berl). 2021 Feb;99(2):289-301. doi: 10.1007/s00109-020-02023-5. Epub 2021 Jan 3. PMID: 33392634.

3. González-Arto M, Aguilar D, Gaspar-Torrubia E, Gallego M, Carvajal-Serna M, Herrera-Marcos LV, Serrano-Blesa E, Hamilton TR, Pérez-Pé R, Muiño-Blanco T, Cebrián-Pérez JA, Casao A. Melatonin MT₁ and MT₂ Receptors in the Ram Reproductive Tract. Int J Mol Sci. 2017 Mar 19;18(3):662. doi: 10.3390/ijms18030662. PMID: 28335493; PMCID: PMC5372674.

Minor:

3) Which tissue is shown in Fig. S1A? What happens in other organs of the mouse, is PGAM5 downregulated in the whole body? The legend for this figure needs to be more detailed.

Explanation

Thanks for your suggestion. We have already revised the legend. Liver tissues form mice were shown in Fig.S1A. :Proteins were isolated from livers. Due to financial constraints, we did not choose to specifically silence PGAM5 in mouse liver. Additionally, we detected PGAM5 expression in mouse kidneys and found that PGAM5 was also silenced in the kidneys (data not shown). 

4) Please check the axis labels in figure 5G: Death cells ...

Explanation

Thanks for your suggestion. We have already revised it.

---

## [Decision Letter · Decision Letter 1]

1 Sep 2024

PONE-D-24-18786R1Melatonin attenuates liver ischemia-reperfusion injury via inhibiting the PGAM5-mPTP pathwayPLOS ONE

Dear Dr. Tang,

Thank you for submitting your manuscript to PLOS ONE. After careful consideration, we feel that it has merit but does not fully meet PLOS ONE’s publication criteria as it currently stands. Therefore, we invite you to submit a revised version of the manuscript that addresses the points still raised by reviewer 2.Please submit your revised manuscript by Oct 16 2024 11:59PM. If you will need more time than this to complete your revisions, please reply to this message or contact the journal office at plosone@plos.org. Please include the following items when submitting your revised manuscript:A rebuttal letter that responds to each point raised by the academic editor and reviewer(s). You should upload this letter as a separate file labeled 'Response to Reviewers'.A marked-up copy of your manuscript that highlights changes made to the original version. You should upload this as a separate file labeled 'Revised Manuscript with Track Changes'.An unmarked version of your revised paper without tracked changes. You should upload this as a separate file labeled 'Manuscript'.

We look forward to receiving your revised manuscript.

Kind regards,

Michael Bader

Academic Editor

PLOS ONE

Reviewers' comments:

Reviewer's Responses to Questions

**Comments to the Author**

1. If the authors have adequately addressed your comments raised in a previous round of review and you feel that this manuscript is now acceptable for publication, you may indicate that here to bypass the “Comments to the Author” section, enter your conflict of interest statement in the “Confidential to Editor” section, and submit your "Accept" recommendation.

Reviewer #2: All comments have been addressed

Reviewer #3: (No Response)

2. Is the manuscript technically sound, and do the data support the conclusions?

Reviewer #2: No

Reviewer #3: Yes

3. Has the statistical analysis been performed appropriately and rigorously? 

Reviewer #2: Yes

Reviewer #3: Yes

4. Have the authors made all data underlying the findings in their manuscript fully available?

Reviewer #2: Yes

Reviewer #3: Yes

5. Is the manuscript presented in an intelligible fashion and written in standard English?

Reviewer #2: Yes

Reviewer #3: Yes

6. Review Comments to the Author

Reviewer #2: (No Response)

Reviewer #3: The authors have not addressed my two major comments:

1) The Scientific Reports paper needs to be cited and the whole manuscript needs to be rewritten telling what is already shown in the published paper and what is new in the current one.

2) The authors need at least to speculate how melatonin can enter the cell and with which receptors it may interact there. Anyhow their data could also be explained by interacting of melatonin with its receptors on the cell surface. There is no evidence that it enters the cell. That's why experiments need to be done on cells with M1 and M2 receptor antagonists to exclude these pathways.

7. PLOS authors have the option to publish the peer review history of their article (what does this mean?). If published, this will include your full peer review and any attached files.

Reviewer #2: No

Reviewer #3: No

---

## [Author Response · Author response to Decision Letter 1]

7 Oct 2024

Reviewer #3: The authors have not addressed my two major comments:

1) The Scientific Reports paper needs to be cited and the whole manuscript needs to be rewritten telling what is already shown in the published paper and what is new in the current one.

Explanation

Thanks very much! The Scientific Reports have been cited. We had two innovations in this study: 1, we identified the role of PGAM5 in melatonin-reducing I/R injury in the liver of mice. 2, Mechanistically, this study extends previous research. There is a certain relationship between the opening of MPTP and mitochondrial fission. The opening of MPTP can trigger the release of calcium ions within the mitochondria and lead to mitochondrial dysfunction, thereby affecting the fission and fusion process of mitochondria. During mitochondrial fission, the opening of MPTP may cause damage to the inner mitochondrial membrane and metabolic instability, which may exacerbate cellular stress and death. Combining our study with this, we can hypothesize that PGAM5-mediated MPTP-opening may be a prerequisite for activating Drp1 phosphorylation and inducing mitochondrial fission. These contents have been integrated into the Instruction and Discussion.

2) The authors need at least to speculate how melatonin can enter the cell and with which receptors it may interact there. Anyhow their data could also be explained by interacting of melatonin with its receptors on the cell surface. There is no evidence that it enters the cell. That's why experiments need to be done on cells with M1 and M2 receptor antagonists to exclude these pathways.

Explanation 

Thanks for your suggestion. We have made the changes according to your suggestion. To investigate which melatonin receptor plays a role, both MT1 and MT2 subtypes were detected in the liver of mice by western blot. Notably, MT2 expression significantly elevated after liver I/R injury, whereas MT1 level remained unaltered (Figure 2 F). Afterward, we used 4P-PDOT to block MT2 receptors. As shown in Figure 2 G and H, 4-P-PDOT negated the effect of melatonin. Thank you again for your proposal.

---

## [Editor Report · Decision Letter 2]

9 Oct 2024

PONE-D-24-18786R2Melatonin attenuates liver ischemia-reperfusion injury via inhibiting the PGAM5-mPTP pathwayPLOS ONE

Dear Dr. Tang,

Thank you for submitting your manuscript to PLOS ONE. After careful consideration, we feel that it has merit but does not fully meet PLOS ONE’s publication criteria as it currently stands. Therefore, we invite you to submit a revised version of the manuscript that addresses the following points: 1) Antibodies agains GPCRs are notoriously unspecific, please provide the details which antibodies you used for MT1 and MT2 and evidence that they are specific.2) Receptors do not need to be upregulated to be involved in physiological actions of ligands. Thus Fig. 2F does not exclude MT1 as mediator of melatonin actions. Please mitigate your conclusion that only MT2 is involved.

We look forward to receiving your revised manuscript.

Kind regards,

Michael Bader

Academic Editor

PLOS ONE
---

## [Author Response · Author response to Decision Letter 2]

9 Oct 2024

Response to Reviewers' Comments

Melatonin attenuates liver ischemia-reperfusion injury via inhibiting the PGAM5-mPTP pathway （PONE-D-24-18786）

1) Antibodies agains GPCRs are notoriously unspecific, please provide the details which antibodies you used for MT1 and MT2 and evidence that they are specific.

Explanation 

Thanks for your suggestion. The details of antibodies have been added to the methods section. 

2) Receptors do not need to be upregulated to be involved in physiological actions of ligands. Thus Fig. 2F does not exclude MT1 as mediator of melatonin actions. Please mitigate your conclusion that only MT2 is involved.

Explanation

Thanks for your suggestion. We have already revised it.

---

## [Editor Report · Decision Letter 3]

15 Oct 2024

Melatonin attenuates liver ischemia-reperfusion injury via inhibiting the PGAM5-mPTP pathway

PONE-D-24-18786R3

Dear Dr. Tang,

We’re pleased to inform you that your manuscript has been judged scientifically suitable for publication and will be formally accepted for publication once it meets all outstanding technical requirements.

Kind regards,

Michael Bader

Academic Editor

PLOS ONE
---

## [Editor Report · Acceptance letter]

18 Oct 2024

PONE-D-24-18786R3 

PLOS ONE

Dear Dr. Tang, 

I'm pleased to inform you that your manuscript has been deemed suitable for publication in PLOS ONE. Congratulations! Your manuscript is now being handed over to our production team.

Kind regards, 

on behalf of

Prof. Michael Bader 

Academic Editor

PLOS ONE